# Time Resolved Polarised Grazing Incidence Neutron Scattering from Composite Materials

**DOI:** 10.3390/polym11030445

**Published:** 2019-03-07

**Authors:** Maximilian Wolff, Apurve Saini, David Simonne, Franz Adlmann, Andrew Nelson

**Affiliations:** 1Department for Physics and Astronomy, Uppsala University, Lägerhyddsvägen 1, 752 37 Uppsala, Sweden; apurve.saini@physics.uu.se (A.S.); davidsimonne@gmail.com (D.S.); franz.adlmann@gmail.com (F.A.); 2ANSTO, Sydney, Lucas Heights, NSW 2234, Australia; anz@ansto.gov.au

**Keywords:** GISANS, composite materials, rheology

## Abstract

Neutron scattering experiments are a unique tool in material science due to their sensitivity to light elements and magnetic induction. However, for kinetic studies the low brilliance at existing sources poses challenges. In the case of periodic excitations these challenges can be overcome by binning the scattering signal according to the excitation state of the sample. To advance into this direction we have performed polarised and time resolved grazing incidence neutron scattering measurements on an aqueous solution of the polymer F127 mixed with magnetic nano-particles. Magnetic nano-composites like this provide magnetically tuneable properties of the polymer crystal as well as magnetic meta-crystals. Even though the grazing incidence small angle scattering and polarised signals are too weak to be evaluated at this stage we demonstrate that such experiments are feasible. Moreover, we show that the intensity of the 111 Bragg peak of the fcc micellar crystal depends on the actual shear rate, with the signal being maximised when the shear rate is lowest (and vice-versa).

## 1. Introduction

Modern technology is rapidly evolving with the development of new and more complex materials. This process is complemented by the design of smaller and smaller functional entities allowing improved energy efficiency and tailored functional properties. With this downscaling the relevance of surface effects is increasing and new experimental capabilities have to be developed. In surface science these include electron microscopy, atomic force microscopy, X-rays, ion beams as well as optical methods. However, due to the strong interaction of the probes with the sample, these methods are mostly limited to the very surface region and are difficult to apply to buried interfaces. Moreover, the detection of light elements as well as magnetic properties on an absolute scale is either not possible or extremely demanding.

Neutrons have a large penetration power for many engineering materials and are also sensitive to light elements as well as to the magnetic induction in materials. An additional advantage is that the scattering cross sections are well known allowing the extraction of information on an absolute scale without the need for complementary calibration measurements. The drawback of neutrons, however, is that they are only available at centralised facilities. Even there, the brilliance and flux of neutrons is limited, in particular, if compared to the latest generation synchrotron radiation facilities.

The above assessment becomes critical for flux demanding methods, like grazing incidence scattering methods or kinetic and time resolved studies, which can yield highly relevant input for a range of scientific as well as technological challenges, like e.g., tribology, directed self-assembly or viscoelasticity. Traditionally, neutron scattering experiments were performed at continuous reactor sources with a single wavelength impinging on a sample. By defining the scattering angle the double differential scattering cross section can be measured with respect to energy and momentum transfer. More recently, the measurement protocol for a range of neutron scattering methods has changed towards time-of-flight methods, using a wide wavelength spectrum and measuring a wider range of momentum and energy transfers at a given time. This applies in particular to neutron reflectometry (NR) that probes nanoscale structure of surfaces and interfaces. The specific advantage in this case is that the reflectivity curve stretches over a wide dynamic range both with respect to momentum transfer (Q) and intensity, with the intensity drop being proportional to Q${}^{-4}$ (Fresnel reflectivity) towards larger Q values above the critical value for total external reflection. Fortunately the spectral distribution of the thermally moderated neutrons is of a Maxwell Boltzmann type (peaking around 0.4–0.5 nm, with a long wavelength tail), which counterbalances the precipitous decrease of intensity; long wavelength neutrons correspond to small Q values, which have a much higher probability of being reflected. It is this wide dynamic range that enables kinetic measurements to take place, since a wide range of Q space can be measured simultaneously. If applied in grazing incidence small angle neutron scattering (GISANS) experiments the approach allows depth resolved experiments [1,2]. In addition, the use of time-of-flight methods can be further optimised by using an intrinsically pulsed source, such as that created by spallation, which is more energy efficient and can offer a higher peak brilliance than continuous reactor sources.

To enable wavelength dispersive experiments the time-of-flight of a neutron from the source to the detector has to be measured and is related to the wavelength. Traditionally this time measurement was done in histograms using the arrival time of a neutron after the trigger from a chopper system. A more recent approach is to store the spatial and time coordinates of every detected neutron in an event file, with the wavelength and momentum transfer being assigned in the post processing. This approach gives much larger flexibility and allows storing meta data, like e.g., temperature or pressure together with every individual neutron. Recently, the feasibility of this approach has been demonstrated [3] and the deformation of the structure of a micellar polymer crystal was followed under oscillatory shear excitation with down to millisecond time resolution [4]. Other, recent experiments, have been performed on a micellar crystal (F127 in EAN) under rheological load [5,6]. In addition, there are studies on worm-like micelles [7,8] and colloidal gels [9] that use strobo-RheoSANS and a generalized protocol to treat the stroboscopic reintegration data has been given [10]. The theoretical limit of this method is very similar to time integrated small angle neutron scattering (TISANE) [11,12] but has the distinct advantage that the signal time integration is done in the post processing and not via histogramming and much less susceptible to systematic errors. Both techniques, our as well as TISANE, have a resolution limit with respect to time which is defined by the neutron pulse length and distance between source, sample and detector [4].

In this manuscript we describe the implementation of time resolved spectroscopic measurements with sub second resolution on the instrument Platypus, ANSTO, Australia [13,14]. Like the previous study probing times down to milliseconds, we use the data acquisition electronics to record the voltage signal from a rheometer to monitor the strain state of the sample, with the voltage file being akin to the neutron event file—each neutron pulse with a corresponding sample state [4]. As distinct to this previous experiment we have combined spectroscopic neutron reflectometry with GISANS [15,16] and polarisation [17].

## 2. Materials and Methods

Figure 1, bottom, shows a photograph of the experimental setup. An Anton Paar MCR500 rheometer (Anton Paar GmbH, Graz, Austria) was mounted at the sample stage of Platypus. In order to apply the oscillatory shear to the sample we used a cone-plate geometry with a cone angle and diameter of 2 degrees and 5 cm, respectively. Due to the geometry of the measurement the sample thickness varies but the shear rate applied to a sample is constant over the whole cone area. We ensured that the mechanical continuum did not break. Hence the resulting velocity over the gap is a linear profile, resulting in a uniform stress rate over the gap, which on the probed surface is sufficient to elucidate discontinuities. The considerations behind this setup lead to the widespread use of this sample geometry for rheological systems with large surface effects. The output voltage, which is proportional to the speed of the measurement cone (1 V corresponds to a rotational velocity in (1/s) and $\frac{\pi }{2}$ phase shifted with respect to the actual strain), is registered at the instruments control computer. Note, the strain has extrema twice per period, whenever the voltage passes zero. This measurement of the voltage allows distributing the neutrons with respect to shear rate or the strain state of the sample. The slit defining the incident beam is visible on the right hand side and the post sample slit in front of the detector entry window to the left. Due to the size of the rheometer it was challenging to mount a sufficient and well defined guide field for the polarised measurements and at this stage we were only able to achieve flipping ratios ($FR=\frac{I_{++}+I_{--}}{I_{+-}+I_{-+}}$, with $I_{++}$, $I_{--}$, $I_{+-}$ and $I_{-+}$ the direct beam intensities of the respective spin channels) of about 5. The overall guide field on Platypus is created by Helmholtz coils and directed along the beam path [17]. To provide a sufficiently high field to define the quantisation axis of the neutrons, perpendicular to the beam path and momentum transfer, at the sample position the field direction is turned adiabatically in the plane of the sample surface. Due to the geometrical restrictions imposed by the rheometer in our case permanent magnets (not shown on the photograph) were used. It turned out that the magnetic stray fields from the permanent magnets did not allow the adiabatic rotation of the field direction and posed particular challenges concerning the polarisation. However, for future studies the optimisation of the guide field is straight forward to implement.

Neutron pulses, as defined by the chopper system, with a repetition rate of 24 Hz (33 Hz for the polarised measurement) enter a block of single crystalline silicon at a narrow edge, see Figure 1 (top), and get scattered at the solid liquid boundary. The wavelength band width was 0.25 to 1.85 nm and 0.25 to 1.3 nm for the unpolarised and polarised experiments, respectively, with a wavelength resolution of 8% defined by the chopper system. The divergence of the incident beam was defined by a slit system. The slit settings were such that the sample was under-illuminated. The angular resolution for an incident beam angle of 0.85${}^{\circ }$ was 3.3%, with a FWHM beam divergence of 0.028${}^{\circ }$ or 0.5 mrad (1.52 mrad for the incident beam setting of 2.5${}^{\circ }$).

Shear is applied via a cone from the top to ensure a constant shear rate throughout the sample. The rheological measurements are performed in situ together with the neutron reflectometry study, similar to setups recently installed at FIGARO, Institute Laue-Langevin (Grenoble, France) [18] or the Liquids reflectometer at the Oak Ridge National Laboratory, TN, USA.

As sample we investigated a 20% in weight solution of the Pluronic F127, received from BASF and used without further purification. The solution was prepared under constant stirring at temperatures of about 5 ${}^{\circ }$C in a liquid micellar phase with deuterated water (effective weight percentage 18.5%) as solvent for better contrast with neutrons. The phase diagram and crystalline structure of this micellar system is known in great detail [19]. For the experimental conditions of our experiment (temperature 28 ${}^{\circ }$C) the sample is in the fcc phase. Close to a solid interface the crystal structure is textured [20]. This texture depends on the termination of the interface as well as on the distance from it [2]. For the present experiment we have used a silicon wafer terminated with native oxide.

Water soluble magnetite nanoparticles with average diameter of (12.3 ± 3.6) nm and a dispersity of 23% were synthesized following the method described in Ref. [21]. Figure 2 shows transmission electron microscope (TEM) images of the particles.

The particle diameter was estimated from their number and the square root of the particle area in the image [22]. In addition to the pure Pluronic solution a 2 mL sample comprising of 20 wt. % (effective weight percentage 18.5% in D${}_{2}$O) Pluronic F127 with dispersed magnetic nanoparticles (3 wt. %) was prepared by thoroughly mixing with a vortex mixer for 15 minutes followed by sonification for 10 min in cold water at 5 ${}^{\circ }$C.

We intended to assemble magnetic nanoparticles by using thermo-reversible block-copolymer cubic crystal formed by Pluronic F127 as a three-dimensional template. The particles are expected to remain trapped in the interstitial cavities of the cubic crystal and form an ordered array since these cavities also follow the crystal lattice. Hence, the crystal structure would consist of an FCC crystal of smaller atoms (particles) intercalated in an FCC crystal of larger atoms (micelles). From previous studies we know that the addition of magnetic nanoparticles influences the crystal quality [23] and the formation and shear alignment of such meta crystals has been shown previously using small angle neutron scattering on silica particles (7 nm diameter) in several Pluronic micellar crystals [24]. Note: The SLD for pluronic F127 is nearly zero ($SLD_{PEO}=0.572\times 10^{-6}$Å${}^{-2}$ and $SLD_{PPO}=0.347\times 10^{-6}$Å${}^{-2}$) [25] but the SLD for Fe${}_{3}$O${}_{4}$ and D${}_{2}$O are very similar $6.91\times 10^{-6}$Å${}^{-2}$ and $6.33\times 10^{-6}$Å${}^{-2}$. This matches the nuclear contrast between the Fe${}_{3}$O${}_{4}$ and D${}_{2}$O and all remaining signal should be magnetic. As a result no asymmetry is expected between the up and down polarised neutrons, since both just measure the difference in SLD, due to the magnetic induction, with respect to the matrix. As the opposite extreme case we matched the Pluronic micelles to a mixture of H${}_{2}$O and D${}_{2}$O to have all the remaining signal from the magnetic particles (data not shown). However, in the respective measurement we were not able to see Bragg reflections.

Figure 2 shows a typical detector image on Platypus. The intensity is plotted as color map versus the neutron wavelength and exit angle. The specular ridge is indicated by the white horizontal line. The (111) reflection of the fcc structure shows up on the specular line for a defined wavelength, corresponding to a momentum transfer along the surface normal of 0.37 nm${}^{-1}$. This strong specular peak is accompanied by diffuse or off-specular scattering related to long range orientational correlations of the micellar crystal in the plane of the solid boundary. From the Q value of the (111) reflection the lattice parameter and the hard sphere diameter (DHS, the distance between adjacent micelle centres) can be calculated to 28.5 nm and DHS = 20.25 nm, respectively. The diameter of the largest spherical entity fitting into an interstitial cavity of a FCC crystal can be calculated geometrically (from the lattice parameter) for octahedral and tetrahedral sites. This value is a rough estimate since the micelles are not hard spheres. The octahedral interstitial site dimension is 8.25 nm but the particle diameter determined from TEM (11.6 nm) is larger and we expect significant interdigitation between segments of nearby coronas and hence defects in the crystal lattice [23].

## 3. Results

We have investigated the near surface structure of the micellar crystal in the vicinity of the silicon wafer terminated by native oxide. Below we provide a systematic summary of the experimental results and discuss them with respect to new instrumental capabilities.

### 3.1. GISANS

Figure 3 depicts the GISANS scattering pattern, integrated over all wavelength, and plotted over the out-of and in-plane exit angle $\alpha_{f}$ and $\alpha_{\left|\right|}$, respectively.

The scattering pattern in the out of plane direction is strongly asymmetric since the beam is reflected for $\alpha_{f}>0$ and transmitted for $\alpha_{f}<0$. These two regions are separated by the sample horizon, indicated by the red line in the panels. The position of the detector defines the scattering angle of the beam, with both the scattering angle and the wavelength defining the momentum transfer. As a result no sharp Bragg reflections, other than the one specularly reflected are visible but the intensity is smeared out along streaks, due to the difference in exit angle for different wavelength fulfilling the Bragg condition for the micellar lattice. At the critical momentum transfer for external reflection the penetration depth into the sample changes by several orders of magnitude and each of the wavelengths (all arriving under the same incident angle $0.85^{\circ }$), probes a different sample volume [2]. The conversion of the signal into Q is additionally complicated by refraction close to the sample horizons and would result in a loss of information but could provide some more details on the texture of the crystalline structure [26]. For further details concerning the scattering geometry we refer to literature [27].

The panel on the top, left depicts data taken for the sample in the crystalline phase without magnetic nano-particles added, and without shear. The well defined streaks in the GISANS signal indicate a relatively well ordered fcc structure. The panel on the top right is for the same sample, while applying a large amplitude oscillatory shear of 1 Hz and 5000%. This clearly reduces the crystallinity of the sample and the streaks in the GISANS signal get less pronounced. The specular and off-specular intensities will be discussed in more detail below.

The lower panels in the figure summarise the scattering patterns taken under identical experimental conditions but now with magnetic nano-particles added to the solution. Under shear a similar trend as for the pure sample is found. However, overall the Bragg reflections visible in the GISANS signal are less pronounced indicating a larger number of defects induced by the magnetic particles [23]. To get an idea on the totally GISANS scattered intensity we have integrated the counts over the detector masking the off-specular intensity. The result is displayed by the numbers $I_{GISANS}$ in the bottom left of each panel. All values are normalised to the measurement of the pure sample without shear. It turns out that with the addition of magnetic particles and/or shear, the GISANS intensity not only gets smeared out but also overall reduced. This result relates to more local defects, scattering to larger Q values for the samples with magnetic particles and/or shear.

Figure 4 depicts detector images taken with a beam of polarised neutrons with a sample with magnetic nano-particles in an applied magnetic field of 100 G, without, top, and under shear, bottom.

As expected, a decrease in the GISANS signal is recorded under the application of oscillatory shear. The detector images taken with up and down polarised neutrons look very similar. This fact supports the hypothesis that the magnetic nano-particles are incorporated in the micellar crystal as defects, rather than sitting at well defined positions in the crystal. This fact will be further supported by the analysis of the specular and off-specular polarised signal provided below. Similar to Figure 3 the intensity values given in the bottom right corner of each panel quantify the GISANS scattering normalised to the sample without shear in this case. Interestingly, under an applied field the GISANS intensity gets reduced more strongly under shear.

### 3.2. Time Resolved Measurements

During the large amplitude oscillatory shear we investigated the intensity of the specular and off-specular intensity of the (111) Bragg reflection. For this purpose we evaluated the output voltage of the rheometer, which is rotational speed (1/s) and proportional to shear rate. The neutron frames defined by the chopper system were sorted with respect to this voltage and the frames for identical voltage ranges were summed together, with the grouped frames then being converted to reflectivity by dividing by a direct beam measurement, Figure 5. For the specular and off-specular signals detector pixels indicated by the white polygon or the region marked as off specular and visible as a green diagonal region in Figure 2 were integrated, respectively. The signals were normalised with respect to the incident flux in the respective voltage interval. Similar to the work of Adlmann et al. [4] we find maxima in intensity whenever the strain on the sample is maximal and the shear rate minimal. It turns out that both the specular as well as the off-specular intensity decreases at higher shear rates and quickly recovers once the shear rate gets lower. The decrease of specular and off-specular intensity under continuous shear and the recovery of it after its cession has been reported before [28]. Note: The specular reflectivity results from scattering length density variations along the normal of the interface, while off-specular scattering originates from long range orientational correlations (correlated roughness of the lattice planes of micelles). As an explanation for both these intensities reducing at the same time several scenarios are possible. The roughness of the layers could become uncorrelated resulting in diffuse scattering away from the Bragg rod. Local defects result in scattering to larger Q values. Tilting of the crystalline structure would limit the interception of the Bragg peak by the Ewald sphere or a reduced texture could result in a less directed scattering. For further details concerning the scattering geometry we refer again to literature [27]. To evaluate these scenarios in more detail more measurements of the off-specular as well as small angle scattering would be needed, which is challenging given the relatively long time needed for one experiment. However, other than in the previous work [4] we did not assign individual rheometer output voltages to each neutron but to frames of neutrons, which reduces the accessible range of time scales. Because of that stroboscopic beating as described in [29] we reach a time resolution of 0.05 to 0.1 seconds as seen in the figure. Sorting frames with respect to the excitation state of the sample has the additional effect of smearing of the data, since neutrons from a frame close to the boundary of the predefined voltage interval may actually scatter from the sample during the next one. Still, as seen we are clearly able to follow the changes in intensity during the excitation cycle.

The panels on the top depict the GISANS scattering patterns sorted with respect to voltage in the same way as the specular and off-specular intensities. Note: The number of neutron frames for a certain voltage interval differs, since the exposure time of the sample in the interval differs. We account for this fact by normalising the scattering patterns with respect to the number of frames in an interval. It turns out that the GISANS intensity integrated over the detector, excluding areas close to $\alpha_{\left|\right|}=0$, is constant over time (see blue data points in the lower panels of Figure 6). However, the GISANS scattering patterns shown in Figure 5 seem to show an intensity variation, which is exactly opposite to the one extracted from the specular and off-specular scattering data. This change in apparent count rate is an effect of the low GISANS signal. For longer exposure times, when the derivative of voltage with respect to time is small, the background on the detector becomes more homogenous, since detector pixels with zero counts show up white in the panels whereas those with at least one count become purple. This leaves the impression of an enhanced GISANS signal for voltage intervals with longer exposure times. In other words the panels on the top panel of Figure 5 depict the GISANS scattering patterns corresponding to different voltages (spindle speeds), in a similar manner to the specular and off-specular intensity variation. While the GISANS images appear to show significant variation in GISANS intensity across the waveform, the integrated GISANS scattering rates show no voltage dependent variation, and the apparent differences are caused by some images being composed of a much larger number of frames than the others. A detailed time dependent analysis would require much longer exposure times. Typically, a GISANS image may take on the order of one hour. Considering the 30 time intervals in our study this would relate to more than one day of measurement but is not a limitation since stronger scattering or more brilliant sources could be used. Still our evaluation of the data clearly shows the capability to evaluate time dependent GISANS scattering patterns with a time resolution below one second.

### 3.3. Polarised Time Resolved Measurements

Figure 6 shows data for the sample containing magnetic nano-particles and using polarised neutrons in the incident beam.

The analysis was done the same way as for the pure sample without polarisation. The specular and off-specular intensities vary in the same way as for the pure samples. Again, similar to the polarised GISANS data (Figure 4) we do not detect a magnetic signal indicating that the magnetic particles do not arrange on an ordered sublattice in the micellar crystal but rather form randomly distributed defects. The lower panels show the variation of the GISANS signal over one period of oscillatory shear excitation. The red line indicates the voltage output signal of the rheometer, which was interpolated previously. No systematic variation of the polarised GISANS signal can be detected but a relatively large scattering of the data points, which is related to counting statistics. This is related to challenges [29] with normalising the data with respect to the incident beam, which was done by dividing the counts by the number of frames for respective voltage intervals.

The study [24] shows data for 3 wt % silica particles (7 nm diameter) dispersed in 25% F127 matrix and a ratio of particles to octahedral interstitial cavities of 0.52 particles per site. We use 3 wt. % magnetite nanoparticles instead, therefore, the magnetic particle concentration is very small and only a fraction of the interstitial sites are filled. At the stoichiometric ratio of one particle per available template site good crystal ordering would be expected. At lower particle concentration the particles should be well templated by the crystal provided that the particle size is smaller than the size of the interstitial cavities and the particles observe at the octahedral or tetrahedral site. At large particle concentrations the particles should form defects.

## 4. Conclusions

In summary, we have combined a periodic excitation on a sample with spectroscopic polarised neutron reflectometry, off-specular scattering and GISANS. Our measurements are flux limited and for a micellar solution with magnetic defects we were not able to extract time resolved data on the magnetisation, or on the GISANS scattering patterns, but it was possible to monitor the specular and off-specular intensity over a duty cycle in large amplitude oscillatory shear with a deformation of 5000% and a frequency of 1 Hz. Longer exposure times will certainly allow the evaluation of the GISANS signal as well as the identification of magnetic contributions. We expect better templating quality for ratios of particle to interstitial sites approaching one, and when the size of the particles are smaller than the voids of the interstitial sites. When these conditions aren’t satisfied the crystallinity of the matrix is perturbed and the correlation between the structure of particles and the templates is reduced. We expect interesting new insights in the functionality of magnetorheological materials by following the approach presented here. For such materials the viscoelastic properties can be tuned by magnetism.

## Figures and Tables

**Figure 1 polymers-11-00445-f001:**
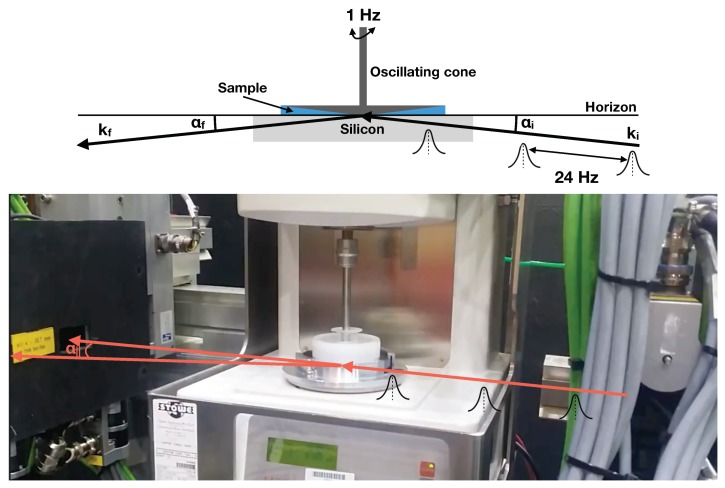
(**Top panel**): Schematics of the experimental setup on Platypus. The pulsed neutron beam (repetition rate 24 Hz) impinges onto the sample while it is under a periodic deformation (periodicity 1 Hz). The incident neutron pulses are illustrated by Gaussians. (**Bottom panel**): Photograph of the experimental setup on Platypus.

**Figure 2 polymers-11-00445-f002:**
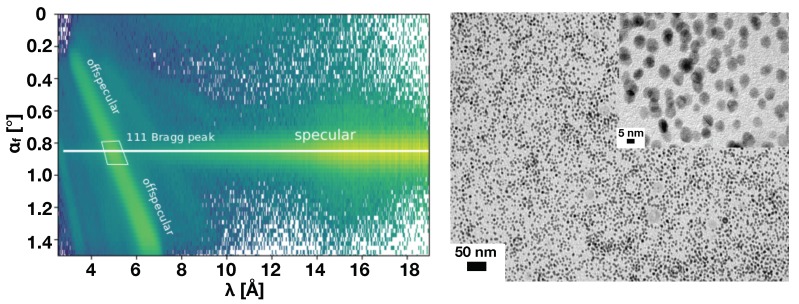
(**Left panel**): Intensity collected at the detector of Platypus and plotted versus out-of-plane exit beam angles and neutron wavelength. The line of specular intensity as well as the 111 reflection of the fcc structure are indicated. (**Right panel**): TEM image of the magnetic particles.

**Figure 3 polymers-11-00445-f003:**
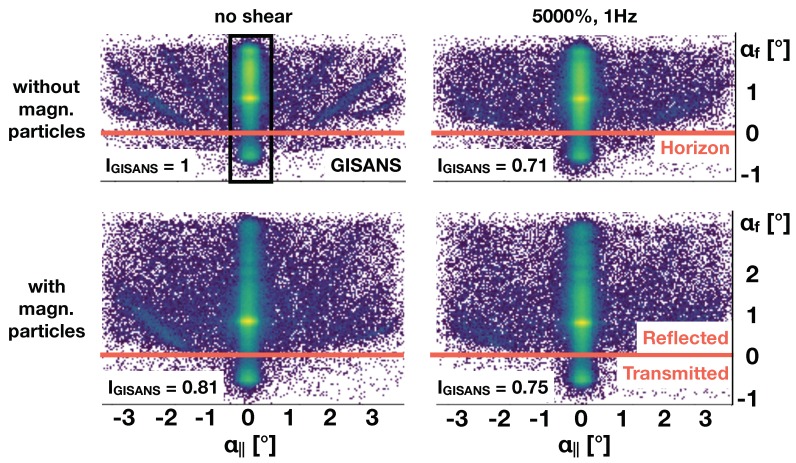
GISANS images of the F127 solution with (**top row**) and without (**bottom row**) magnetic particles taken with (**right column**) and without (**left column**) oscillatory shear. Data is summed over all wavelengths and plotted versus in and out-of-plane exit angle. The shear as well as the addition of magnetic particles makes the GISANS scattering more diffuse. The relative GISANS intensities ($I_{GISANS}$ given at the bottom left of each panel) were calculated by integrating a detector region with non-zero in-plane scattering angle (i.e., excluding specular/off-specular region) and normalising by the number of frames contributing towards each image.

**Figure 4 polymers-11-00445-f004:**
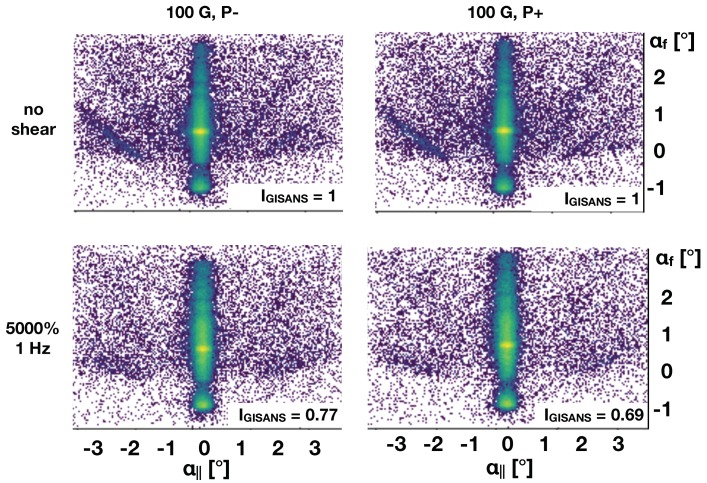
Polarised GISANS images of the F127 solution with magnetic particles taken without (**top row**) and with oscillatory shear (**bottom row**). Data taken for the different polarisation directions of the incident neutron beam are shown in the left and right column. Shear smears out the GISANS scattering. The relative GISANS intensities ($I_{GISANS}$ given at the bottom right of each panel) were calculated by integrating a detector region with non-zero in-plane scattering angle (i.e., excluding specular/off-specular region) and normalising by the number of frames contributing towards each image.

**Figure 5 polymers-11-00445-f005:**
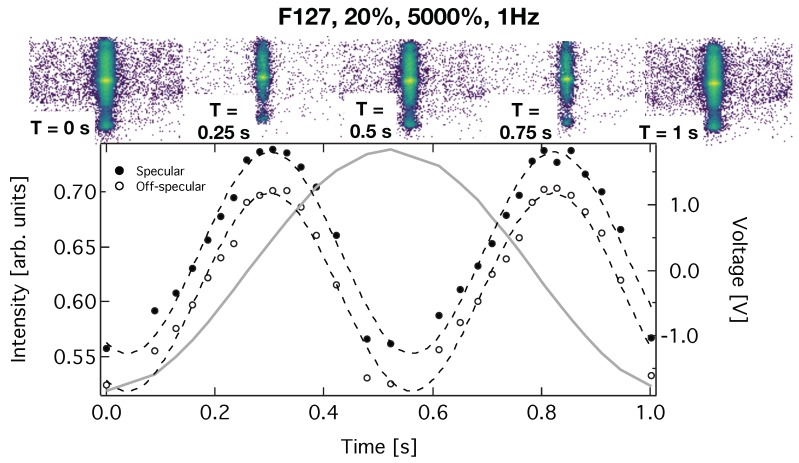
The panels on the top show GISANS detector images extracted during different time intervals (indicated in the lower left corner of each panel) during the oscillatory shear. The aparent intensity variations in the signal are related to counting statistics and have no physical meaning. The closed and open circles in the lower panel are the specular and off-specular intensities integrated over the specular region indicated by the polygon and the Bragg sheet visible as diagonal green area of increased intensity in Figure 2, respectively, and normalised by the number of frames contributing towards each image. The dashed line represents a sin wave fitted to the data. The solid grey line shows the output voltage (right axis) of the rheometer, which is proportional to the shear rate. A maximum in scattering intensity is visible for the smallest shear rates.

**Figure 6 polymers-11-00445-f006:**
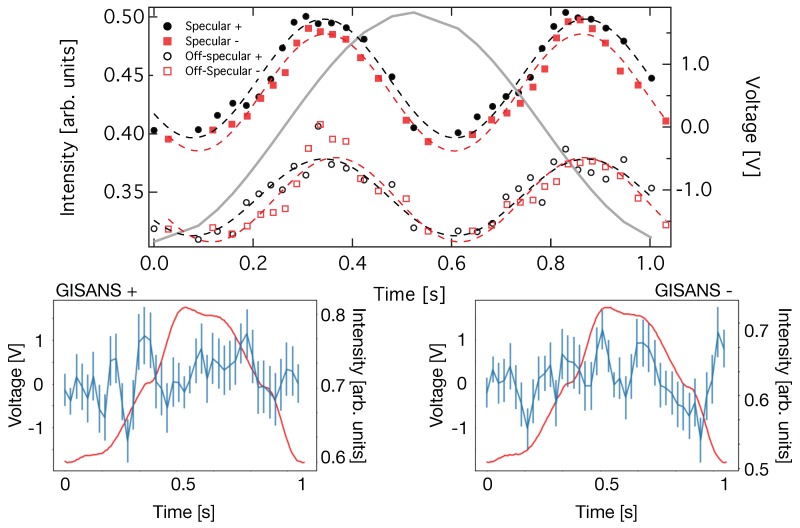
Neutron intensity plotted versus time during one period of oscillation. (**Upper panel**): The closed symbols represent intensity integrated over the specular region indicated by the polygon in Figure 2 and normalising by the number of frames contributing towards each image. The black and red color indicates scattering for + and − polarised incident neutrons. The open symbols represent the off-specular scattering extracted in the same way but integrating over the Bragg sheet visible as green diagonal region of increased intensity in Figure 2. (**Lower panels**): GISANSsignal calculated by integrating a detector region with non-zero in-plane scattering angle (i.e., excluding specular/off-specular region) and normalising by the number of frames contributing towards each image and plotted versus time. The left and right panels show data for + and − polarised incident neutrons, respectively. The solid red line is the output voltage of the rheometer.

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
