# Peer review of "Time Resolved Polarised Grazing Incidence Neutron Scattering from Composite Materials"

_polymers, 2019, doi:10.3390/polym11030445_

Round 1

Reviewer 1 Report

Dear Editor,

I have reviewed the manuscript “Time resolved polarised grazing incidence neutron scattering from composite materials” by Max Wolff, Apurve Saini, David Simonne, Franz Adlmann and Andrew Nelson. Topic discussed in the manuscript is quite important and timely. Authors are developing a novel approach for time-resolved neutron scattering experiments. Authors claim that they can measure neutron scattering cross sections with the time resolution of order of 0.1 sec. However, in my opinion this experimental manuscript lacks of many important details. Authors should provide all relevant information and demonstrate their results in a more clear way. Therefore, I do not recommend publishing the manuscript in the current from. Below, please, find my questions that should be addressed prior my recommendation.

1)      Can author provide more details on the experimental scheme. The upper panel in Fig. 1 is not informative enough. Can author show where the sample is (is it a blue region?), where the Si (probably light gray?) substrate is. How do they apply stress (rotation of the dark grey cone?). What is the thickness of the sample?

2)      The shear strain is applied to the top surface of the sample. Neutron beam impinges on the sample from the bottom. Does the strain applied on the top influences the bottom surface? What is the strain spatial distribution? Is it homogeneous across the thickness of the sample? Is the strain homogeneous along the radius of the cone?  I would expect that higher strain appear at higher radius. Can authors explain this in more details?

3)      In Fig. 3 the information on angles should be provided. Where is the plane of incidence? Why the scattering is not symmetric in the out of plane direction? While authors refer to their earlier paper it would be helpful if they provide the scattering geometry (showing all angles shown in Fig. 3) in this manuscript as well.

4)      Fig. 5 is confusing. The GISANS images are plotted as a function of voltage, while the bottom panel shows the intensity as a function of time. Due to the counting problems discussed by the authors these dependencies are opposite which is confusing. May be it is more relevant to present GISANS images not for voltage intervals but for time intervals. This data is more relevant to the experiment.

I would say that the images in fig. 5 does not present the physical results but just results of incorrect data processing.

Can authors explicitly explain how they calculate the intensity function in Fig. 5. Do they integrate data in each image over (for example) the specular region and then average this intensity over several consecutive frames within a given time interval? I cannot believe that taking the data from the lower panel of Fig. 6 one can get the smooth periodic function.

Is the gray line in Fig. 5 a voltage as a function of time? How does this correlate with the red solid line in Fig. 6?

5)      Can authors elaborate on the dependence of the strain on voltage. It seems that voltage oscillation period is twice smaller than that of strain. Please, clarify this.

6)      Magnetic anisotropy axes of particles are mostly randomly oriented and moreover the particles are mostly in superparamagnetic state, meaning that magnetization fluctuates. The applied magnetic field is rather weak and cannot fully magnetize the particles. Why authors expect to obtain spin-dependent signal in their experiments? Can authors extend discussion of magnetic properties of their system.

7)      That would be great is authors in their conclusion (or discussion section) compare their technique to other time-resolved techniques mentioned in the introduction (TISANE, stroboscopic SANS, kinetic SANS). Do they see any way to get higher time resolution than in TISANE for example.

Author Response

Dear Editor,

we would like to thank the referees for their detailed and constructive feedback on our manuscript. We are happy to hear that the instrumental developments presented by us are generally judged interesting and worth publishing. We agree with the overall judgment of the reviewers that the initial manuscript was lacking some clarity with respect to data presentation and interpretation. For the revised version of the manuscript we have carefully gone through the manuscript and improved the clarity wherever needed. We hope that the manuscript is now judged suitable for publication.

I have reviewed the manuscript “Time resolved polarised grazing incidence neutron scattering from composite materials” by Max Wolff, Apurve Saini, David Simonne, Franz Adlmann and Andrew Nelson. Topic discussed in the manuscript is quite important and timely. Authors are developing a novel approach for time-resolved neutron scattering experiments. Authors claim that they can measure neutron scattering cross sections with the time resolution of order of 0.1 sec. However, in my opinion this experimental manuscript lacks of many important details. Authors should provide all relevant information and demonstrate their results in a more clear way. Therefore, I do not recommend publishing the manuscript in the current from. Below, please, find my questions that should be addressed prior my recommendation.

1)      Can author provide more details on the experimental scheme. The upper panel in Fig. 1 is not informative enough. Can author show where the sample is (is it a blue region?), where the Si (probably light gray?) substrate is. How do they apply stress (rotation of the dark grey cone?). What is the thickness of the sample?

We have included labels in the figure to indicate the sample, silicon and cone.

Moreover, we have added the sentences: In order to apply the oscillatory shear to the sample we used a cone plate geometry with a cone angle and diameter of 2 degrees and 5 cm, respectively.  Due to the geometry of the measurement the sample thickness varies but the shear rate applied to a sample is constant with cone radius.

2)      The shear strain is applied to the top surface of the sample. Neutron beam impinges on the sample from the bottom. Does the strain applied on the top influences the bottom surface? What is the strain spatial distribution? Is it homogeneous across the thickness of the sample? Is the strain homogeneous along the radius of the cone?  I would expect that higher strain appear at higher radius. Can authors explain this in more details?

We have added the following section:

Due to the geometry of the measurement the sample thickness varies but the shear rate applied to a sample is constant with cone radius. The velocity profile in the measurement gap is linear resulting in a constant shear rate and strain.

3)      In Fig. 3 the information on angles should be provided. Where is the plane of incidence? Why the scattering is not symmetric in the out of plane direction? While authors refer to their earlier paper it would be helpful if they provide the scattering geometry (showing all angles shown in Fig. 3) in this manuscript as well.

We now define the respective angles in Fig. 1, label the axis and give the exit beam angles on the graphs.

We have added the following sentences explaining the asymmetry of the out-of plane scattering:

The scattering pattern in the out of plane direction is strongly asymmetric since the beam is reflected for $\alpha_f > 0$ and transmitted for $\alpha_f < 0$.

These two regions are separated by the sample horizon, indicated by the red line in the panels.

4)      Fig. 5 is confusing. The GISANS images are plotted as a function of voltage, while the bottom panel shows the intensity as a function of time. Due to the counting problems discussed by the authors these dependencies are opposite which is confusing. May be it is more relevant to present GISANS images not for voltage intervals but for time intervals. This data is more relevant to the experiment.

The rheometer output is a voltage signal, which is proportional to the shear rate in the sample. Since the excitation is not linear equal intervals in voltage will not result in equal time intervals. To clarify this we give the time stamps of the GISANS pictures now in Fig. 5.

I would say that the images in fig. 5 does not present the physical results but just results of incorrect data processing.

We agree with the referee that the apparent changes in count rate in the GISANS signal are not related to physics. They are related to the difference in time the detector is exposed to neutrons for the individual voltage intervals. We take this effect into account when normalising to the incident neutron flux and find a constant GISANS intensity. With the images we try to illustrate challenges when looking on data with low count rates as well as the capability of extracting the GISANS images for short time intervals.

To clarify this we have rewritten the corresponding paragraph in the revised version of the manuscript.

Can authors explicitly explain how they calculate the intensity function in Fig. 5. Do they integrate data in each image over (for example) the specular region and then average this intensity over several consecutive frames within a given time interval? I cannot believe that taking the data from the lower panel of Fig. 6 one can get the smooth periodic function.

We have added the sentences: For the specular and off-specular signals detector pixels indicated by the white polygon or the region marked as off specular in Fig. \ref{RTOF} were integrated, respectively. After integration the signals were normalised with respect to the incident flux by dividing by the number of frames in the respective voltage interval.

Concerning the intensity variation we can only comment that this is what we get following our procedure. At this moment we are not in the position to discuss the intensity variations from the physics point of few, since we would need complementary SANS as well as GISANS data. However, the continuous variation of the signal may well be physical assuming a reorientation or change in crystal size under deformation.

Is the gray line in Fig. 5 a voltage as a function of time? How does this correlate with the red solid line in Fig. 6?

The grey line is an interpolation. We explain this in the revised version of the manuscript:

The red line indicates the voltage output signal of the rheometer, which was interpolated previously.

5)      Can authors elaborate on the dependence of the strain on voltage. It seems that voltage oscillation period is twice smaller than that of strain. Please, clarify this.

We have added the following clarification: Note, the strain has extrema twice per period, whenever the voltage passes zero.

Generally, the strain has the same period but changes sign during one oscillation.

6)      Magnetic anisotropy axes of particles are mostly randomly oriented and moreover the particles are mostly in superparamagnetic state, meaning that magnetization fluctuates. The applied magnetic field is rather weak and cannot fully magnetize the particles. Why authors expect to obtain spin-dependent signal in their experiments? Can authors extend discussion of magnetic properties of their system.

Actually, our particles are at the limit of paramagnetism and ferro-magnetism. Still in a liquid they will behave super paramagnetic. At this stage we are focusing on showing the capability in doing polarised measurements but this will need further optimisation. During the experiment presented here the polarisation on the beam line was relatively bad and therefore the field at the sample position was low. Since we were not able to measure an effect at this stage a detailed discussion of magnetic ordering goes beyond the scope of this technical manuscript.

7)      That would be great is authors in their conclusion (or discussion section) compare their technique to other time-resolved techniques mentioned in the introduction (TISANE, stroboscopic SANS, kinetic SANS). Do they see any way to get higher time resolution than in TISANE for example.

TISANE as well as stroboscopic reintegration with reach a time resolution defined only by the source and geometrical parameters concerning the layout of the instrument. The distinct advantage of our method is that it is more robust since the data are histogrammed in the post processing.

We have added the sentence and provide one more citation:

Both techniques, our as well as TISANE, have a resolution limit with respect to time which is defined by the neutron pulse length and distance between source, sample and detector.

Reviewer 2 Report

The authors have made some important steps in the process towards time resolved structural measurements of composite magnetic materials with complex methods. So in principle it is interesting for colleagues in the field to read about the progress to get more ideas for their own approach towards developing methods like these. From that point of view I am positive about this manuscript.

The quality of the presentation needs some severe improvements. Some of the figures are of very bad quality. Most of the captions are incomplete, which makes it impossible to interpret the results. The interpretation of the results is unclear.

This has to be significantly improved to consider it for publication.

Some details:

Line 66: I assume that “subtle” should be “susceptible”.

Line 82: it would be good to define the flipping ratio for the outsiders.

Figure 1: None of the symbols, the pulses nor the 24 Hz in the top panel are explained in the caption.

Line 96: How is it possible that when the sample is under-illuminated it acts as a slit? Please explain this in the text.

Figure 2 right panel: what is on the vertical axis? I cannot read it.

Figure 2 left panel: the text with the size bar is much too small to read.

Figure 3: the caption is not describing what is seen in the graph. Please move the explanation in the text to the caption.  Now the caption and text aren’t even consistent.

Figure 4: the caption is incomplete.

Line 179-181: I don’t understand that the fact supports the hypothesis. I don’t see anything that could break the asymmetry between the up and down polarisation. Please give a better and more complete explanation.

Figure 5: This is an unclear graph. It needs a good descriptive caption. The data points should have error bars. Some lines are drawn in the graph. What does each line represent? Is it a fit or a direct calculation. On the right axis a voltage is plotted. Wouldn’t it be better to have a more significant unit? For example the applied shear rate?

Figure 5 interpretation: how is it possible that both the specular and off-specular intensity simultaneously go up and down? Where do the neutrons go to? You observe a maximum intensity at 0 voltage, what does that mean. Does 0 voltage correspond to low or high shear? At the same time the image in the top panel shows a high intensity. What is wrong?

Lines 200-213: This is completely unclear to me. Are the authors describing an instrumental artefact they don’t understand. Or do they understand it and don’t they correct for it.  

Figure 6: error bars are missing. Not explanation is given of the meaning of which symbol/line reflects which quantity. Specially with the lower panel this is a problem.  

The results are not discussed in an understandable manner. I have the feeling that the authors do have an explanation for their observation, but it doesn’t get across to me.

Author Response

Dear Editor,

we would like to thank the referees for their detailed and constructive feedback on our manuscript. We are happy to hear that the instrumental developments presented by us are generally judged interesting and worth publishing. We agree with the overall judgment of the reviewers that the initial manuscript was lacking some clarity with respect to data presentation and interpretation. For the revised version of the manuscript we have carefully gone through the manuscript and improved the clarity wherever needed. We hope that the manuscript is now judged suitable for publication.

The authors have made some important steps in the process towards time resolved structural measurements of composite magnetic materials with complex methods. So in principle it is interesting for colleagues in the field to read about the progress to get more ideas for their own approach towards developing methods like these. From that point of view I am positive about this manuscript.

The quality of the presentation needs some severe improvements. Some of the figures are of very bad quality. Most of the captions are incomplete, which makes it impossible to interpret the results. The interpretation of the results is unclear.

This has to be significantly improved to consider it for publication.

All figure captions have been rewritten to improve the clarity.

The interpretation of the data has been rewritten following the comments from reviewer 1.

Some details:

Line 66: I assume that “subtle” should be “susceptible”.

Fixed

Line 82: it would be good to define the flipping ratio for the outsiders.

We included the definition.

Figure 1: None of the symbols, the pulses nor the 24 Hz in the top panel are explained in the caption.

The information has been added to the caption.

Line 96: How is it possible that when the sample is under-illuminated it acts as a slit? Please explain this in the text.

We thank the referee for spotting this mistake, which has been corrected in the revised version.

Figure 2 right panel: what is on the vertical axis? I cannot read it.

The readability of the labels was improved.

Figure 2 left panel: the text with the size bar is much too small to read.

The readability of the labels was improved.

Figure 3: the caption is not describing what is seen in the graph. Please move the explanation in the text to the caption.  Now the caption and text aren’t even consistent.

The caption has been changes accordingly.

Figure 4: the caption is incomplete.

The caption has been expanded.

Line 179-181: I don’t understand that the fact supports the hypothesis. I don’t see anything that could break the asymmetry between the up and down polarisation. Please give a better and more complete explanation.

As explained in the manuscript the formation of a magnetic sublattice should result in an asymmetry of the polarised neutron signal.

Figure 5: This is an unclear graph. It needs a good descriptive caption. The data points should have error bars. Some lines are drawn in the graph. What does each line represent? Is it a fit or a direct calculation. On the right axis a voltage is plotted. Wouldn’t it be better to have a more significant unit? For example the applied shear rate?

The caption has been expanded. The error bars can be estimated from the scattering of points. Otherwise it is not entirely trivial to calculate exact error bars, since for a limited number of frames the spectrum of the incident beam, which can not be measured simultaneously becomes relevant for the uncertainties.

Figure 5 interpretation: how is it possible that both the specular and off-specular intensity simultaneously go up and down? Where do the neutrons go to? You observe a maximum intensity at 0 voltage, what does that mean. Does 0 voltage correspond to low or high shear? At the same time the image in the top panel shows a high intensity. What is wrong?

We have rewritten this part and hope it is more clear now.

Lines 200-213: This is completely unclear to me. Are the authors describing an instrumental artefact they don’t understand. Or do they understand it and don’t they correct for it.

We have rewritten this part and hope it is more clear now.

Figure 6: error bars are missing.

Error bars have been added.

Not explanation is given of the meaning of which symbol/line reflects which quantity. Specially with the lower panel this is a problem.

The caption has been expanded.

The results are not discussed in an understandable manner. I have the feeling that the authors do have an explanation for their observation, but it doesn’t get across to me.

This comment is difficult to address since it is not specific. We have rewritten most of the interpretation as well as we have improved the quality of the figures and the captions. We Hope that this clarifies our interpretation of the data and will be happy to answer more detailed questions from the reviewer if needed.

Reviewer 3 Report

This manuscript describes about the solution structure of Pluronic F127 coexisting with magnetic nano-particles at the solid-liquid interface by means of TOF-GISANS and NR. By the combination of TOF-neutron scattering and application of oscillatory shear during the scattering experiments, the authors were able to perform time-resolved measurements with a time resolution of 0.05 to 0.1 seconds. Furthermore, the statistics of the experimental results were much improved by sorting the scattering intensities based on the monitored shear rate by making use of the event data recordings. In addition, polarized neutron scattering technique was also applied in order to reveal the magnetic order of the mixed nano-particles. All the attempts in the report are original, so that I have no objection to the publication on this conference proceedings.

On the other hand, I have a few comments and questions about the contents:

 In the caption of Fig. 2, “Right” and “Left” are the other way aroun

 In Fig. 3, “Off-specular” below the top left graph should be removed.

 The data shown in Fig. 3 and Fig. 4 seem to be raw records on the detectors. If it is so, the data should be converted to normalized intensity as a function of Q.

I will be happy if the authors respond to these.

Author Response

Dear Editor,

we would like to thank the referees for their detailed and constructive feedback on our manuscript. We are happy to hear that the instrumental developments presented by us are generally judged interesting and worth publishing. We agree with the overall judgment of the reviewers that the initial manuscript was lacking some clarity with respect to data presentation and interpretation. For the revised version of the manuscript we have carefully gone through the manuscript and improved the clarity wherever needed. We hope that the manuscript is now judged suitable for publication.

This manuscript describes about the solution structure of Pluronic F127 coexisting with magnetic nano-particles at the solid-liquid interface by means of TOF-GISANS and NR. By the combination of TOF-neutron scattering and application of oscillatory shear during the scattering experiments, the authors were able to perform time-resolved measurements with a time resolution of 0.05 to 0.1 seconds. Furthermore, the statistics of the experimental results were much improved by sorting the scattering intensities based on the monitored shear rate by making use of the event data recordings. In addition, polarized neutron scattering technique was also applied in order to reveal the magnetic order of the mixed nano-particles. All the attempts in the report are original, so that I have no objection to the publication on this conference proceedings.

On the other hand, I have a few comments and questions about the contents:

 In the caption of Fig. 2, “Right” and “Left” are the other way around?

We have corrected this.

 In Fig. 3, “Off-specular” below the top left graph should be removed.

Off-specular has been removed.

 The data shown in Fig. 3 and Fig. 4 seem to be raw records on the detectors. If it is so, the data should be converted to normalized intensity as a function of Q.

I will be happy if the authors respond to these.

Yes the data in Fig. 3 and 4 are raw detector images. Due to refraction as well as the large changes in penetration depth of different wavelength the conversion to Q can result in misleading results. A detailed analysis goes beyond the scope of this more technical paper but is certainly of high interest for future studies. To clarify this we have expanded the respective text in the revised version of the manuscript.

Round 2

Reviewer 1 Report

Dear Editor,

Authors reworked their manuscript and improve essentially the quality of presentation. Now the manuscript can be published.

Reviewer 2 Report

I am happy with the revisions made to the article. It is now much better accessible to the reader.